# Outstanding Contributions of LAL Technology to Pharmaceutical and Medical Science: Review of Methods, Progress, Challenges, and Future Perspectives in Early Detection and Management of Bacterial Infections and Invasive Fungal Diseases

**DOI:** 10.3390/biomedicines9050536

**Published:** 2021-05-11

**Authors:** Hiroshi Tamura, Johannes Reich, Isao Nagaoka

**Affiliations:** 1LPS Consulting Office, Tokyo 160-0023, Japan; 2Department of Host Defense and Biochemical Research, Graduate School of Medicine, Juntendo University, Tokyo 113-8421, Japan; nagaokai@juntendo.ac.jp; 3Microcoat Biotechnologie GmbH, 82347 Bernried, Germany; j.reich@microcoat.de

**Keywords:** endotoxin, LPS, LAL test, sepsis, (1→3)-β-D-glucan, fungal infections

## Abstract

The blue blood of the horseshoe crab is a natural, irreplaceable, and precious resource that is highly valued by the biomedical industry. The Limulus amebocyte lysate (LAL) obtained from horseshoe crab blood cells functions as a surprisingly sophisticated sensing system that allows for the extremely sensitive detection of bacterial and fungal cell-wall components. Notably, LAL tests have markedly contributed to the quality control of pharmaceutical drugs and medical devices as successful alternatives to the rabbit pyrogen test. Furthermore, LAL-based endotoxin and (1→3)-β-D-glucan (β-glucan) assay techniques are expected to have optimal use as effective biomarkers, serving as adjuncts in the diagnosis of bacterial sepsis and fungal infections. The innovative β-glucan assay has substantially contributed to the early diagnosis and management of invasive fungal diseases; however, the clinical significance of the endotoxin assay remains unclear and is challenging to elucidate. Many obstacles need to be overcome to enhance the analytical sensitivity and clinical performance of the LAL assay in detecting circulating levels of endotoxin in human blood. Additionally, there are complex interactions between endotoxin molecules and blood components that are attributable to the unique physicochemical properties of lipopolysaccharide (LPS). In this regard, while exploring the potential of new LPS-sensing technologies, a novel platform for the ultrasensitive detection of blood endotoxin will enable a reappraisal of the LAL assay for the highly sensitive and reliable detection of endotoxemia.

## 1. Introduction

It has been more than five decades since the discovery of the remarkable benefits of horseshoe crab blood in the rapid detection of bacterial components [1]. LAL is an aqueous extract of horseshoe crab (*Limulus polyphemus*) blood cells, and the LAL test is the most sensitive and reliable method applied for in vitro detection of bacterial endotoxins [2]. Endotoxins, also known as LPS, derived from the cell membrane of Gram-negative bacteria, are closely linked to biological properties such as pyrogenicity, toxicity, and mitogenicity and are involved in the production of proinflammatory cytokines through Toll-like receptor (TLR) 4 signaling. The gel-clot LAL test was approved by the Food and Drug Administration (FDA) in the 1970s and has been widely adopted as the official method for detecting bacterial endotoxins [3]. Iwanaga et al. were the first to report that LAL is activated by bacterial endotoxin based on the endotoxin-triggered factor-C (FC)-mediated coagulation cascade [4]. LAL is also activated by a major polysaccharide from the fungal cell wall, β-glucan, via the β-glucan-activated factor-G (FG)-mediated coagulation pathway (Figure 1) [5]. Based on these findings, we successfully developed the world’s first endotoxin and β-glucan-specific chromogenic LAL reagents (Endospecy and Gluspecy; Seikagaku Corp., Tokyo, Japan) [6,7]. Gluspecy is a former trademark of G-test. For the purpose of evaluating endotoxin contamination in parenteral drugs, medical devices, and raw materials, three types of LAL tests (gel-clot, chromogenic, and turbidimetric techniques) were harmonized in 2012 for compendial methods listed in the three pharmacopeias: The United States Pharmacopeia (USP), Japanese Pharmacopeia (JP), and European Pharmacopeia (Ph. Eur.).

However, to date, the clinical application of the LAL test as a useful biomarker of systemic endotoxemia in septic conditions has not been successful. In Japan, chromogenic and turbidimetric techniques with endotoxin-specific LAL assays after appropriate pretreatment have been used extensively since their approval by the Ministry of Health, Labour and Welfare (MHLW, Tokyo, Japan) [8]; however, there are several unresolved technical issues related to plasma extraction methods, the physical and biological properties of endotoxin circulating in the blood, and the enzymatic degradation of endotoxin molecules, and these limitations have negative impacts on the early intervention for patients at risk for severe sepsis [9]. By contrast, the β-glucan-specific assay in human blood (Fungitec G-test (Seikagaku Corp.), Fungitell (Associates of Cape Cod., Inc. (ACC), East Falmouth, MA, USA) has been proven to be highly effective as a rapid, accurate adjunct in the clinical diagnosis of invasive fungal diseases [10,11]. Consequently, approval by the FDA lay the foundation for adopting the serum β-glucan assay as a global standard for the early diagnosis of invasive fungal diseases.

In this review, we first focus on the development history, recent advances, and limitations of the LAL assay, as well as plasma extraction methods and potential methodologies that may notably improve the LAL technique. Next, we discuss possible solutions to the above-mentioned issues and future development perspectives for blood endotoxin detection in sepsis and septic shock. Finally, we introduce the diagnostic performance of the serum β-glucan assay and its contribution to the early diagnosis of patients at risk for invasive fungal diseases and fungal septicemia.

## 2. History of the LAL Test and Its Clinical Application

In 1964, Levin and Bang made the novel discovery that the lysate from Atlantic horseshoe crab (*L. polyphemus*) coagulated upon contact with bacterial endotoxins, and their epoch-making research findings contributed to the remarkable development of the LAL test. In 1977, the FDA approved the LAL test reagent (gel-clot), which was initially launched by ACC In 1983, Seikagaku Corp. developed an innovative LAL assay using a chromogenic substrate, which evolved into highly sensitive LAL assays based on a series of pioneering studies by Iwanaga et al. Notably, the LAL test is the most sensitive, specific, and quantitative method among various physicochemical, immunological, and biological techniques for detecting LPS. In this context, we successfully developed a new technology platform coupled with instrumentation and software, laying the foundation for a new paradigm for differentiating between endotoxin and β-glucan [6]. This not only led to more specific and reliable quality control tests for pharmaceutical products but also provided a scheme for the adjunct diagnosis of Gram-negative bacterial (GNB) sepsis or invasive fungal infections.

Rapid diagnostic methods for detecting microorganisms have been increasingly important because blood cultures remain the gold standard for the microbiological diagnosis of bacterial and fungal infections, despite the fact that the techniques are rather time-consuming and have low sensitivity, particularly after antibiotic initiation. Non-culture-based LAL techniques for detecting endotoxin were considered beneficial for the early diagnosis and screening of GNB infection. However, LAL methods have not been highly endorsed, as their clinical efficacy in patients with GNB infections remains unclear and unsatisfactory because endotoxin levels determined by the LAL assay do not always reflect the clinical status or severity of diseases [12]. The probable reason for this is that a number of studies were undertaken using conventional LAL tests that are not specific only to endotoxins. In addition, the test results depend on the sensitivity and interference susceptibility of LAL and how the pre-treatments of blood samples were performed. Furthermore, the timing of specimen collection for blood cultures and the time until start LAL testing as well as bacterial species can affect the outcome of the tests.

## 3. Gram-Negative Bacteremia and Endotoxemia

Gram-negative bacteria are the most frequently isolated microorganisms and can cause bacteremia. The research evidence reveals that bacteria can release LPS directly into the human blood stream, as reported by Brandtzaeg et al. [13]. Antibiotic treatment with ceftazidime and imipenem may increase plasma endotoxin levels in patients presenting with positive blood cultures for Gram-negative bacteria [14]. Thus, it is important to understand the concordance between endotoxemia and Gram-negative bacteremia [15]. Yoshida et al. reported that the sensitivity and specificity of Endospecy for systemic GNB infections were 69.7% and 96.3%, respectively, but Gram-negative bacteria grew in only 39.7% of endotoxemia cases [8]. A meta-analysis by Hurley et al. revealed that the association was surprisingly weak, particularly for Gram-negative bacteremia caused by *Escherichia coli* [12]. In addition to the different types of Gram-negative bacteremia, blood pretreatment techniques, as well as the sensitivity and the specificity of LAL, are regarded as critically important considerations for the appropriate interpretation of the poor concordance.

## 4. Factors in Blood Samples That Interfere with the Limulus Test

Before conducting the LAL assay, the plasma sample needs to be pretreated to prevent interference with the LAL coagulation cascade, which consists of serine protease zymogens (Factor C, Factor B (FB), and proclotting enzyme (pro-CE)). There are three types of interferents in human blood. The first is serine proteases, including trypsin, factor X, and α-thrombin, which are capable of hydrolyzing native or synthetic chromogenic substrates (Coagulogen, Boc-Leu-Gly-Arg-p-nitroanilide), leading to false-positive results. The second is serine protease inhibitors, such as trypsin inhibitor and antithrombin III, which can cause false-negative results by inhibiting coagulation enzymes. The third is substances capable of binding to LPS: if LPS forms complexes with LPS-binding proteins or peptides, it is unable to trigger Factor C, leading to the inactivation of LAL activity. These substances include amphiphilic and mostly cationic polypeptides, such as polymyxin, LL-37, defensin, and bacterial/permeability-increasing protein (BPI), which are antimicrobial (host defense) peptides [16,17,18].

Another effect of interference is endotoxin masking, resulting in issues such as low endotoxin recovery (LER), which is caused by the dissociation of the supramolecule assembly of LPS in the presence of surfactants and chelating agents. The phenomenon of LER was initially reported in the therapeutic products of monoclonal antibodies in April 2013, and the FDA has considered LER to be a potential safety issue owing to the probability of false-negative test results. During the LER phenomenon, a common formulation matrix containing sodium citrate and polysorbate in biopharmaceuticals results in the inability to recover LPS in a time-dependent manner when spiked into undiluted samples [19,20]. It is not entirely clear whether masked endotoxins are biologically active in vivo, although masked endotoxin is reported to be a potent trigger of immune responses [21]. It is interesting to note that a similar phenomenon in clinical specimens is assumed to occur during disaggregated LPS–HDL binding, for instance [22].

## 5. Pretreatments for Overcoming Interferents in Blood

Numerous parenteral drug products can be used for the LAL test after dilution with endotoxin-free distilled water or an aqueous solution on the condition that dilutions not exceed the maximum valid dilution. In 1973, Nachum et al. evaluated cerebrospinal fluid (CSF), which has extremely low protein content compared to blood, using the LAL test as a potential tool to detect GNB meningitis. This study indicated that positive LAL tests in CSF were observed in 38 (100%) of 38 culture-proven GNB meningitis cases [23]. CSF is ideal for the LAL test because of its relatively low content of interfering substances derived from blood. In several subsequent studies, the LAL test was proven to be a sensitive, efficient, and accurate diagnostic tool for detecting GNB meningitis. Inagaki et al. first demonstrated that the normal concentration of CSF endotoxin was <3 pg/mL using Endospecy, and the concentrations found in neonates were the same as those found in infants [24]. Patients with GNB meningitis were found to have markedly elevated levels of endotoxin.

In contrast, clinical specimens such as plasma, serum, and whole blood cannot be prepared for the LAL test by dilution alone. These samples need to be heated at high temperatures after dilution or be exposed to chloroform, acetic acid, perchloric acid, nitric acid, potassium hydroxide, and surfactants prior to the LAL endotoxin reaction. These pretreatments have been more effective in allowing the appropriate recovery of endotoxin from LPS-spiked specimens. Numerous clinical studies based on different pretreatment methods have been conducted since Levin’s pioneering investigation [12,15,16,17,23,24,25,26,27,28,29,30,31,32,33]. Based on a large number of papers, Novitsky comprehensively reviewed the related literature [27], including our studies, and demonstrated that the use of whole blood samples made the LAL test an ideal method for improving the convenience and efficiency of the endotoxin assay [30]. A whole blood assay, however, still poses significant challenges, primarily due to a higher degree of interference. Consequently, platelet-rich plasma has been used instead of ordinary plasma for the effective detection of endotoxin.

In the US, there is still no clear evidence that shows that endotoxin released from bloodborne bacteria can act as an efficient biomarker for the early diagnosis of Gram-negative sepsis. The primary reason for this is that most ambiguous results were obtained from conventional LAL tests that react to β-glucan rather than endotoxins [31]; in addition, host LPS inactivation mechanisms and interactions of LPS with humoral and cellular components in peripheral blood may affect LAL results. Thus, the difficulty in establishing the usefulness of the LAL assay for Gram-negative sepsis remains a serious barrier to its regulatory acceptance by the FDA. The FDA has not yet accepted the LAL test for blood endotoxin detection, as it has been quite difficult to clarify the clinical significance of the behavior of circulating endotoxins. In particular, the LAL test used in several studies performed in the US and Europe contained conventional formulations and was not endotoxin-specific. In Japan, as stated above, the chromogenic method (Toxicolor: conventional LAL; Endospecy: endotoxin-specific LAL) was the first approved in vitro diagnostic (IVD) for patients with GNB infection [26,32]. After that, the manufacturing and marketing of these products were discontinued because they did not meet physician expectations. Consequently, the kinetic turbidimetric assay (ES test WAKO; FUJIFILM Wako Pure Chemical Corp., Osaka, Japan) became the only approved IVD product [33], in spite of the fact that it often cannot detect endotoxin, even in patients with septic shock, because the clinical sensitivity remains very low [34].

## 6. Improvement of LAL Technology

Given the above challenges, endotoxin scattering photometry (ESP), a novel LAL assay, was developed to markedly improve the sensitivity by adopting a laser light-scattering particle-counting method. ESP is characterized by the method based on the same principle of turbidimetric assay; however, ESP can detect very small particles in a fluid generated in the reaction, which is the first appearance of LAL cascade triggered by endotoxin. Shimizu et al. showed that the ESP assay was even more useful than previous tests for detecting endotoxin, depending on the severity of septic patients [35]; nevertheless, this technique has not been authorized by MHLW for widespread use. Furthermore, chromogenic LAL coupled with a bioluminescence technique has been newly developed using mutant firefly luciferase (Luminutes-ET; DKK-TOA, Tokyo, Japan) [36]. The unique technique is built on their research findings that a mutant North American firefly (*P. pyralis*) luciferase generates luminescence intensity from enzymatic luminescence reaction more than 10-fold higher than that of wild-type luciferase. This assay enables highly sensitive quantitation of endotoxin; however, its clinical application has not been attempted beyond its use in monitoring endotoxins in dialysate to ensure high-quality and safe hemodialysis.

Novel analytical techniques (i.e., Pep-Abser (Peptide Door Co. Ltd., Takasaki, Japan) and EndoLISA (Hyglos GmbH, a bioMérieux company, Bernried, Germany)) have been developed in which endotoxin in aqueous samples is first adsorbed to a solid phase coated with endotoxin-binding molecules, followed by the elimination of interfering substances and the subsequent analysis of endotoxin levels using the LAL assay [37,38]. The assays are usually based on high-affinity binding through the originally developed protein and peptide coated onto microplate wells. Very few studies on this enzyme-linked immunosorbent assay (ELISA)-like format address the sensitivity improvements in the blood endotoxin assay. However, this kind of technology is highly expected to efficiently measure and evaluate trace amounts of blood endotoxin.

## 7. Various Techniques Involving the LAL Assay and Other Methods

In 1972, the LAL test was widely used in the pharmaceutical and medical industries. Since then, the LAL test has been extensively evaluated as an extremely sensitive, specific, simple, rapid, and economical method to detect endotoxins. Various alternative techniques have also been developed without using LAL technology since the 1970s.

As shown in Figure 2, endotoxin assay can be divided into two categories: LAL and non-LAL assay. The LAL assay is officially used with a different type of formulation that comprises conventional or endotoxin-specific reagents for both end-point and kinetic assay formats [6,39]. Other techniques (modified LAL) include ones such as ESP, the bioluminescence assay using mutant luciferase, and the ELISA-like assay as described earlier. As a different approach, a lab-on-a-chip device (Endosafe-PTS; Charles River Laboratories, Charleston, SC, USA) was first introduced as a modified technique to further improve the usability and simplicity of the LAL assay [40]. Furthermore, LAL alternatives based on recombinant technologies have recently been attracting a great deal of attention from the perspective of the global pharmacopoeia [41,42,43,44]. Recombinant alternatives are specific to endotoxin, and consist of two types of reagents: recombinant Factor C and cascade enzymes (see “next-generation LAL technology”). Recently, Bolden et al. reviewed currently available recombinant alternatives to horseshoe crab blood lysates and their comparability [45].

Among non-LAL based assays, GC/MS determination aims to target lipid A and its 3-hydroxy fatty acid molecules. Some studies suggest that GC/MS analysis could be clinically effective if its sensitivity and accuracy were markedly improved [9,46]. A quantitative method was developed by fluorescence labeling of 3-hydroxy fatty acid, but a complex process with preparative HPLC is required [47]. ELISA is a simple and specific method and could be used as a diagnostic tool if its sensitivity were significantly enhanced and if the system were able to recognize various LPS molecules [48]. For GC/MS and ELISA, the technical difficulties in making them suitable for practical use in the laboratory do not appear to be easy to overcome. LPS capture methods can offer a beneficial approach for horseshoe crab conservation, while these depend on the extent to which LPS binding molecules recognize diverse structures of LPS [49]. LPS O-antigen has been successfully targeted to detect LPS using polyclonal and monoclonal antibodies against the O-antigen. However, it has often yielded results that are inconsistent with those of the LAL assay [50].

## 8. Fluorescence Spectroscopy and Electrochemistry-Based Rapid Determinations

Fluorescence spectroscopy has been widely explored since the 1990s. The first reported quantification of endotoxin was based on the use of fluorescence-labeled recombinant endotoxin-neutralizing protein (rENP) and a fluorescence polarization (FP) instrument (Polar Scan, ACC; no longer available) [51]. The FP technique enables the rapid detection of endotoxin at concentrations as low as 0.2 EU/mL, although the application has mostly been limited to environmental samples. Takano et al. reported that the electrochemical signal accompanied by the chromogenic LAL assay using differential pulse voltammetry (DPV) was markedly enhanced and able to detect endotoxin in the range of 0.5–1000 EU/L [52]. Bai et al. reported the development of an electrochemical aptasensor (an aptamer-based biosensor) with cascade signal amplification by three-way DNA junction-aided enzymatic recycling and graphene nanohybrids for amplification. This technique can provide an ultrasensitive LPS assay to the femtogram level (10 fg/mL) [53]. These biosensor-based technologies can provide important insights for the development of an extremely sensitive and rapid endotoxin assay in blood and its application in point-of-care testing (POCT) devices, which are available for bedside testing and monitoring for septic patients.

## 9. Indirect Assay for Endotoxin

Cell-based endotoxin assays were developed using different immune cells and cell lines, such as human neutrophils, monocytes, and human embryonic kidney (HEK) 293 cells. An alternative in vitro pyrogen test, the monocyte activation test (MAT), has been developed to detect non-endotoxin pyrogen (NEP) as well as endotoxins [54]. HEK 293 clones expressing TLR4-MD2-CD14 were used to detect endotoxin based on LPS-triggered NF-kB activation [55]. MAT and engineered HEK techniques are beneficial tools for determining biologically active endotoxin molecules capable of producing proinflammatory cytokines. Hiki et al. revealed the potential utilization of the HEK technique to adequately assess endotoxemia caused by the release of biologically active endotoxin from Gram-negative bacteria accompanied by antibiotic action [56], but no reports are available on the diagnostic applications of the MAT. However, using the MAT has demonstrated the clinical importance of NEP [57].

Based on a different mechanism, the endotoxin activity assay (EAA) measures the production of reactive oxygen species by human neutrophils from whole blood subjects, followed by the formation of LPS–anti-LPS-antibody complexes. Following Marshall and Walker’s initial research on an observational cohort, the EAA (Spectral Medical Inc.; Toronto, Canada) received 510(k) de novo clearance from the FDA in 2003, in addition to procalcitonin (PCT), in order to help identify patients at an increased risk of developing sepsis after ICU admission [58]. EAA has been globally adopted to evaluate the efficacy of polymyxin B hemoperfusion (PMX-HP using Toraymyxin (Toray Industries, Inc., Tokyo, Japan)) in patients enrolled in clinical trials [59]. Importantly, measuring EAA levels can be rather useful for identifying patients expected to benefit the most from PMX-HP therapy. With regard to the correlation between EAA and the LAL assay, patients with Gram-positive bacterial infections were reported to have higher EAA values than controls and those with GNB infections, and there was no significant correlation between the two [60]. It is apparent that different mechanisms are responsible for these inconsistent results, and thus, further studies might be needed to clarify their clinical relevance. As for analytical validation of cell-based assay, the mechanism of the above-mentioned HEK Blue LPS detection (InvivoGen, San Diego, CA, USA) is based on LPS-induced activation of NF-κB in HEK293 cells. Accordingly, this assay allows the detection of biologically active endotoxin with high specificity. In contrast, MAT and EAA have low or limited specificity against endotoxin due to the mechanism by which the analytes are generated on a series of cellular responses.

## 10. Relevant Analytes

In addition to the above-mentioned techniques, several methodologies known as indirect endotoxin assays are available. These include measurements of serum lipoproteins, antiendotoxin antibody, and LBP. Levels et al. revealed that the levels of high-density lipoprotein (HDL) typically decrease in the plasma and lymph of septic individuals [61]; however, LPS binds mainly to low-density lipoprotein (LDL) and very-low-density lipoprotein (VLD) in hypertriglyceridemic serum from septic patients [62]. Using an inhibition ELISA, Allan et al. demonstrated that anti-bacteroides lipopolysaccharide IgG levels markedly fluctuated compared with those in the sera of healthy subjects [63]. Interestingly, an exploratory study conducted by Opal et al. indicated that LBP levels were elevated in 97% of sepsis patients compared with normal control values [64]. According to Opal’s study, no correlation was found between endotoxemia and LBP levels, and very little information is available regarding the correlation among the different markers, including sCD14. Nonetheless, circulating levels of LBP and soluble CD14 are recognized as clinical biomarkers of endotoxemia, and thus it is expected that it will be informative for understanding the severity of sepsis and septic shock. Furthermore, indirect assays could also be useful for better understanding the pathological condition during illness in combination with other biomarkers, such as PCT, sCD14, and interleukin-6 [65]. These assays provide abundant information to clinical investigators and physicians on various pathophysiological states in septic patients and are diagnostic strategies for the effective management of sepsis. All methods described in the Section 7, Section 8 and Section 9 are summarized in Table 1.

## 11. Low-Grade Endotoxemia Related to Chronic Inflammation

Bacterial translocation occurs more frequently in patients with intestinal mucosal damage and in immunocompromised hosts and may lead to septic conditions. It may also occur in healthy individuals without any severe effects, but little information is available on the long-term consequences. The term “low-grade endotoxemia” or “metabolic endotoxemia” was recently coined for a better representation of subclinically elevated levels of circulating blood endotoxin. It has been speculated that circulating endotoxins, usually attributed to the translocation of gut-microbiota-derived LPS from the gastrointestinal tract into the bloodstream, can cause or exacerbate chronic inflammatory diseases, to which inflammation-related diseases such as ischemic heart disease, stroke, cancer, diabetes mellitus, chronic kidney disease, nonalcoholic fatty liver disease (NAFLD), and autoimmune and neurodegenerative conditions are attributable [66,67,68,69]. Several studies have reported the relationship between low-grade endotoxemia and metabolic diseases such as insulin resistance and obesity [66,70,71]. The problem is that the majority of the results were obtained from the conventional LAL assay [25], which reacts to both endotoxin and β-glucan. Replacement of the LAL test used in these studies with the endotoxin-specific LAL assay should make the data more reliable and valid. Additionally, an alternative method using ELISA for the detection of low-grade endotoxemia can be used to determine gut permeability and bacterial translocation, although its sensitivity and specificity relative to the LAL assay are unknown [69]. More interestingly, Finkelman et al. suggested that the detection of (1→3)-β-D-glucanemia rather than endotoxemia may serve as a novel biomarker for fungal translocation into systemic circulation and improve the understanding of leaky gut syndrome [72].

## 12. Next-Generation LAL Technology

In the 1990s, Muta and Iwanaga successfully cloned the cDNA of Japanese horseshoe crab coagulation factors. Based on these findings, Ding et al. of the National University of Singapore developed a new endotoxin-specific assay (PyroGene; Lonza, Walkersville, MD, USA) introduced by Cambrex (now Lonza) using only recombinant factor C from Southeast Asian horseshoe crab (*Carcinoscorpius rotundicauda*) [41]. Two German companies, Hyglos GmbH and Haemochrom Diagnostica GmbH (Essen, Germany) launched rFC-based endotoxin assay kits comparable to PyroGene using different species of horseshoe crab. Thereafter, a novel chromogenic LAL reagent (PyroSmart) containing all of the recombinant factors from horseshoe crab and a chromogenic substrate, Boc-Leu-Gly-Arg-pNA, was successfully created by Seikagaku Corp. and ACC (Pyrosmart NextGen) [42]. The rFC assay and the cascade reagent described in the General Information of the Japanese Pharmacopeia Eighteenth Edition (Draft) have been effective as of 1 April 2021.

In the case of Ph. Eur., a new general chapter on the test for bacterial endotoxins using recombinant factor C (2.6.32) was published in July 2020. For USP, the proposed chapter “<1085.1> Use of Recombinant Reagents in the Bacterial Endotoxins Test” was released in PF 46(5) in 2020. These recombinant reagents may have some considerable advantages over natural-resource-dependent LAL technology from the perspective of horseshoe crab conservation. In addition, recombinant reagents are more consistent since they are not subject to the lot-to-lot variability usually found in LAL [43]. While clinical application of rFC and recombinant cascade reagents have not yet been carried out using human blood, these products might provide valuable tools for the detection of endotoxin, particularly in the context of the needs for POCT [73].

## 13. Contribution of LAL-Based β-Glucan Assay in Invasive Fungal Diseases

Invasive fungal diseases are an increasingly common etiology of sepsis in severely ill patients, resulting in high morbidity and mortality [74]. *Candida* and *Aspergillus* spp. are the most prevalent nosocomial fungal pathogens. Some yeast agents, including *Candida albicans*, *Candida tropicalis*, and *Candida glabrata*, are reported to be the most common causative agents, followed by *Aspergillus* and non-*Aspergillus* molds. The most common fungi causing invasive infections in patients with hematological malignancies are *Aspergillus* spp. Early diagnosis is highly effective in treating invasive fungal diseases; however, it was particularly challenging until the late 1990s because fungal cultures were associated with markedly low positivity rates and were time-consuming. Successful development of non-culture-based diagnostic techniques, such as β-glucan and galactomannan in blood, was a remarkable breakthrough that provided great benefits for the early diagnosis of invasive fungal diseases [10,75]. As summarized in Table 2, a multicenter cohort study published in *The Lancet* by Obayashi et al. revealed that a quantitative β-glucan assay in blood using β-glucan-specific chromogenic LAL reagents (Fungitec G-test) is highly efficient in the diagnosis and management of invasive fungal diseases, with 90% sensitivity, 100% specificity, and 97% negative predictive value [10]. After that, β-Glucan test Wako (LAL-based turbidimetric assay; FUJIFILM Wako Pure Chemical Corp.) became available in Japan. In addition, Fungitec G-test MK (kinetic version) has already been licensed to Nissui Pharmaceutical Co., Ltd., Tokyo, Japan, and kept on delivering with the product name, Fungitec G-test MK-II.

Key opinion leaders in the US and Europe expressed their immense interest in Obayashi’s pioneering achievement, and thus, the technology transfer for manufacturing the β-glucan assay kit was successfully undertaken in partnership between Seikagaku Corp. and Associates Cape Cod, Inc., in the US in 2004. Based on the results of a multicenter clinical evaluation by Ostrosky et al. [11], the US product Fungitell became the first and only FDA-cleared in vitro diagnostic test for invasive fungal diseases.

## 14. Steadily Increased Use of Fungitell in Global Diagnostic Laboratories

Fungal diagnosis using β-glucan was included in the diagnostic criteria for probable invasive fungal infections in the revised European Organization for Research and Treatment of Cancer/Invasive Fungal Infections Cooperative Group and the National Institute of Allergy and Infectious Diseases Mycoses Study Group (EORTC/MSG) [80]. New aspergillosis guidelines from the Infectious Diseases Society of America (IDSA) recommend serum and bronchoalveolar lavage galactomannan as a marker for the diagnosis of invasive *Aspergillus* (IA) in high-risk hematologic malignancy and allogeneic HSCT patients. Serum β-glucan assays are also recommended for diagnosing IA, although these tests are not specific for the infection. The guidelines from IDSA and the European Society for Clinical Microbiology recommend considering the non-culture-based fungal test for the detection of invasive candidiasis and aspergillosis. Furthermore, appropriate diagnostic tests should be useful for assessing whether empiric antifungal therapy can be safely discontinued to avoid the unnecessary use of antifungal agents. Importantly, the β-glucan level in the presence of systemic antifungal therapy is not decreased, even in patients receiving >7 days of treatment. According to the latest review by Finkelman, the diagnostic performance of β-glucan assays has been assessed in approximately 200 publications, including multiple meta-analyses. The heterogeneity of many studies is relatively high, but diagnostic performance characteristics are reasonably reproducible [72]. Systematic reviews and meta-analyses on β-glucan assays present satisfactory results, reporting a sensitivity of 76–83% (91 facilities) and a specificity of 79–85% (91 facilities) [76,77,78,79] (Table 2). In addition, it should be noted that β-glucan levels have to be carefully interpreted before making a diagnosis because of the expected occurrence of false-positive results [6].

Recently, an ELISA-based β-glucan assay (Immunotesta β-glucan; Sekisui Medical Co., Ltd., Tokyo, Japan) [81] was approved by MHLW on the basis of a strong correlation with the data obtained from the LAL-based assay. This approach may not only help to reduce false-positive rates but also be incorporated into POCT devices, although benchmark and application studies are needed. For almost three decades, the LAL-based β-glucan assay has been practiced as a powerful adjunct to the diagnosis of invasive fungal diseases. Moreover, it may be helpful as a biomarker for the diagnosis of *Pneumocystis jiroveci* pneumonia (PJP) in patients with HIV [82]. Fungitell has been adopted as a diagnostic tool in numerous countries in conjunction with the dissemination of guidelines. In addition to FDA clearance, Fungitell is CE marked and thus available in the US and all European Union countries.

## 15. Future Perspective

The LAL assay has established a firm position as an alternative to the rabbit pyrogen test, and thus, the horseshoe crab has already proven to be an extremely beneficial organism for biomedical use. However, there is growing awareness of the importance of protecting endangered species, and thus, alternative assay technologies using recombinant LAL have gained attention with data accumulation [45,83]. As stated earlier, clinical evaluation of the LAL assay remains challenging for a number of reasons, including various interferents and the emerging issue of endotoxin masking. Consequently, improved techniques would be especially useful in demasking and capturing LPS molecules in circulating blood. Thus, new approaches might be helpful in detecting the presence of trace amounts of endotoxin in the blood and properly evaluating the clinical effect of direct hemoperfusion and the therapeutic potential of new drug candidates, such as anti-endotoxin agents and antimicrobial peptides [84].

On another front, with the successful development of a simple and rapid automatic method for the detection of β-glucan [85], in addition to a lab-on-a-chip device or a lateral-flow device in ready-to-use format and potential applications of factor-G-derived β-glucan binding site, beta-glucan-binding protein of LAL, and beta-glucan recognition proteins (insect BGRPs) [4,40,86,87,88], the diagnostic value of the blood β-glucan assay will continue to grow in the future, with potential widespread use in clinical laboratories. Given the increasing need for POCTs for infectious diseases, a novel blood endotoxin assay platform based on recombinant LAL and microflow devices would also represent a breakthrough in the early diagnosis, prognosis, and treatment monitoring of sepsis and septic shock.

## Figures and Tables

**Figure 1 biomedicines-09-00536-f001:**
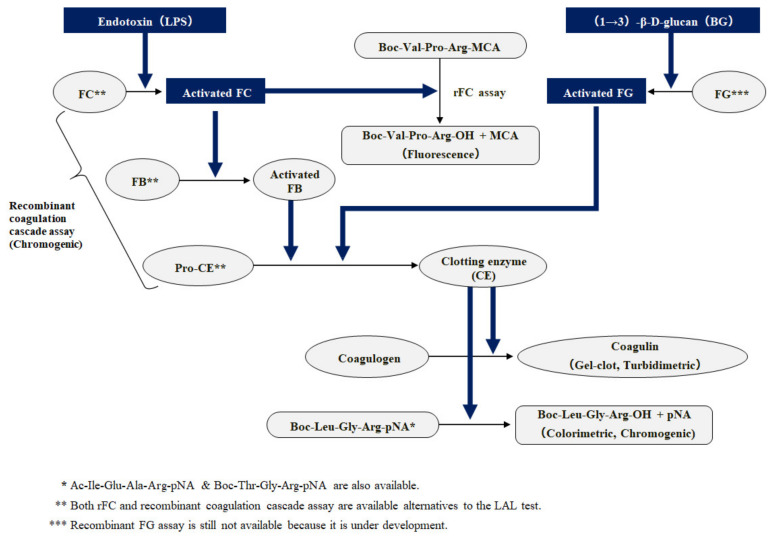
Coagulation cascade in Limulus amebocyte lysate.

**Figure 2 biomedicines-09-00536-f002:**
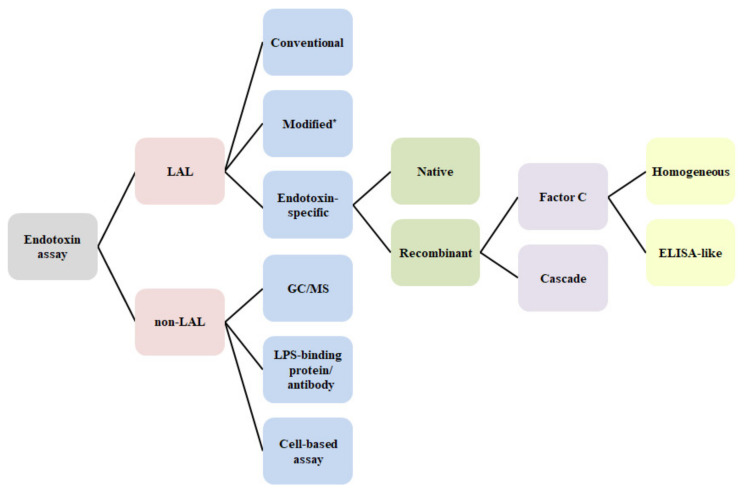
Different types and formats of endotoxin assays. * Lab-on-a-chip (disposable cartridge); Target LPS capture (ELISA-like); Endotoxin Scattering Photometry (ESP); Bioluminescence assay using mutant luciferase; Electrochemical LAL assay.

**Table 1 biomedicines-09-00536-t001:** Current techniques and potential methods of endotoxin detection in clinical and pharmaceutical samples.

LAL/	Analyte	Technique	Principle/Key Elements	Method	Sample	Reference
Non-LAL						
LAL	Endotoxin	Conventional/ Endotoxin-specific	Activation of pro-clotting enzyme in Limulus amebocyte lysate	Gel-clot	Plasma	Levin, J. [25]
				Chromogenic/Turbidimetric	Plasma	van Deventer, S.J. [15] Tamura, H. [16,17] Obayashi, T. [26] Inada, K. [28] Kambayashi, J. [33]
		LAL alternatives (Endotoxin-specific)	Recombinant Factor C	Fluorogenic	Non-clinical	Ding, J.L. [41]
			Recombinant coagulation enzymes	Chromogenic	Pharmaceutical Non-clinical	Mizumura, T. [42]
		Modified LAL	Endotoxin scattering photometry (ESP)	Light scattering by small particles	Plasma	Shimizu, T. [35]
			Engineered firefly luciferases with improved sensitivity	Chromogenic	Dialysate	Noda, K. [36]
			Target LPS capture with LPS binding peptide produced by phage-display	Chromogenic	Pharmaceutical Non-clinical	Suzuki, M.M. [37]
			Target LPS capture with phage-derived protein	Chromogenic/ Fluorogenic	Pharmaceutical Non-clinical	Grallert, H. [38]
			Endosafe PTS device with a disposable cartridge	Chromogenic	Nuclear medicine, Pharmaceutical	Maule, J. [40]
			Electrochemical LAL assay	Chromogenic	Dialysate	Takano, S. [52]
Non-LAL (Direct)	Endotoxin/	GC/MS	β-hydroxymyristic acid content of Lipid A	Gas chromatograph-Mass spectrometer	Serum (Rabbit)	Maitra, S.K. [46]
	Lipid A	Immunological techniques	Antiserum to J5 mutant of E.coli 0111:B4	ELISA	Milk	Mohammed, A.H. [48]
	Endotoxin		Target LPS capture with polymyxin probe	ELISA	Non-clinical	Inoue, K.Y. [49]
	LPS O-antigen		Antiserum to O-polysaccharides	Radioimmuno-assay	Non-clinical	Munford, R.S. [50]
	Endotoxin	Fluorescence spectroscopy	Fluorescence-labelled Endotoxin Neutralizing Protein (ENP)	Fluorescence polarization	Non-clinical	Sloyer, J. [51]
		Electro-chemistry-based	Electrochemical aptasensor	Signal amplification by enzymatic recycling	Non-clinical	Bai, L.J. [53]
Non-LAL (Indirect, Cell-based)	Endotoxin	NF-κB	Monocyte activation test	Cytokines production	Pharmaceutical	Hoffman, S. [54]
	Pyrogen	MAT	Toll-like receptor 4/MD-2/CD14	NFkB- reporter assay	Plasma (Rat)	Nishida, M. [56]
	Endotoxin activity	EAA	Human neutrophil-complement opsonized LPS-IgM complexes	Chemilumi-nescent emission	Whole blood	Marshall, J.C. [58]
Relevant analytes	Anti-endotoxin antibody	Immunological techniques	Anti-bacteroides lipopolysaccharide IgG	Inhibition ELISA	Serum	Allan, E. [63]
	LBP		Endotoxin-LBP-sCD14 complexes	ELISA	Serum	Opal, S.M. [64]

**Table 2 biomedicines-09-00536-t002:** Diagnostic performance of the β-glucan assay for the detection of invasive fungal diseases.

Clinical Papers	Sensitivity (%)	Specificity (%)	Evidence		β-Glucan Assay Kits
Obayashi, T., et al. [10]	90	100	Multi-center clinical studies, 9 sites	Japan	Fungitec G-test
Ostrosky-Zeichner, L., et al. [11]	70	87	Multi-center clinical evaluation, 6 sites	USA	Fungitell
Lu, Y., et al. [76]	76	85		13 **	Fungitell (9/13)
					WAKO (2/13)
					Fungitec G-test (1/13)
Karageorgopoulos, D.E., et al. [77]	77	85		23 **	Fungitell (10/23)
					Fungitec G-test (7/23)
					WAKO (5/23)
			Meta-analysis *		Gold Mountain River (1/23)
Onishi, A., et al. [78]	80	82		36 **	Fungitell (17/36)
					Fungitec G-test (9/36) WAKO (6/36)
					Gold Mountain River (3/36) Others (1/36)
White, S.K., et al. [79]	83	79		19 **	Fungitell (19/19)

* Multicenter cohort study, Multicenter case-control study, Prospective cohort study, Prospective case-control study, Retrospective cohort study, Retrospective case-control study, etc. ** Total number of articles.

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
