# Peer review of "Outstanding Contributions of LAL Technology to Pharmaceutical and Medical Science: Review of Methods, Progress, Challenges, and Future Perspectives in Early Detection and Management of Bacterial Infections and Invasive Fungal Diseases"

_biomedicines, 2021, doi:10.3390/biomedicines9050536_

Round 1
Reviewer 1 Report
In this manuscript, Tamura et al reviewed the endotoxin and B-glucan detection methods, particularly LAL assay. The authors gave a historical perspective, methods as well as its application. Overall the review is very informative and well written. However, I have the following comments.
1- The authors mentioned that LAL assay do not reflect the clinical status and severity of the disease. It would be more informative if the authors outline the probable reasons of this phenomenon.
2- The section-6 about improvement of LAL technology is very short. Please expand this section explaining the mechanism in detail.
3-The authors mentioned indirect cell based assay such as Nf-kB, MAT and EAA. Activation of NfkB and MAT is a general mechanism of many signaling processes. The authors need to describe the specificity of these assays to determine the endotoxin levels.
4- Is there any correlation between the levels of CD14 ,LBP and endotoxemia. Will it be useful to diagnose the severity based on multiple parameters?
5-Please incorporate the reference for " several studies have reported......and obesity (Line 347-348).
Author Response
Dear reviewer #1:
We really appreciate your positive feedback, and thoughtful comments that helped improve the manuscript.
Our point-by-point responses are as follows.
Comment 1:
The authors mentioned that LAL assay do not reflect the clinical status and severity of the disease. It would be more informative if the authors outline the probable reasons of this phenomenon.
Response:
Thank you very much for your helpful suggestion. We have supplemented section 2 with explanations of the probable reasons as follows.
The probable reason for this is that a number of studies were undertaken using conventional LAL tests that are not specific only to endotoxins. In addition, the test results depend on the sensitivity and interference susceptibility of LAL and how the pretreatments of blood samples were performed. Furthermore, timing of specimen collection for blood cultures and the time until start LAL testing as well as bacterial species can affect the outcome of the tests.
Comment 2:
The section-6 about improvement of LAL technology is very short. Please expand this section explaining the mechanism in detail.
Response:
According to your advice, we have supplemented the section 6 with explanations of the mechanism as follows.
- ESP: ESP is characterized by the method based on the same principle of turbidimetric assay; however, ESP can detect very small particles in a fluid generated in the reaction, which is the first appearance of LAL cascade triggered by endotoxin.
- Luminitz-ET: The unique technique is built on their research findings that a mutant North American firefly (P. pyralis) luciferase generates luminescence intensity from enzymatic luminescence reaction more than 10-fold higher than that of wild-type luciferase.
Comment 3:
The authors mentioned indirect cell based assay such as Nf-kB, MAT and EAA. Activation of NfkB and MAT is a general mechanism of many signaling processes. The authors need to describe the specificity of these assays to determine the endotoxin levels.
Response:
We appreciate your query. We understand that's an important point. In the section 2, we have provided the following explanation of the specificity of these assays.
As for analytical validation of cell-based assay, the mechanism of above-mentioned HEK Blue LPS detection is based on LPS-induced activation of NF-κB in HEK293 cells. Accordingly, this assay allows the detection of biological active endotoxin with high specificity. In contrast, MAT and EAA have low or limited specificity against endotoxin due to the mechanism by which the analytes are generated on a series of cellular responses.
Comment 4:
Is there any correlation between the levels of CD14 ,LBP and endotoxemia. Will it be useful to diagnose the severity based on multiple parameters?
Response:
You have raised an interesting question. We made a comment on this matter in the section as follows.
According to Opal’s study, no correlation was found between endotoxemia and LBP levels. However, very little information is available regarding the correlation among the different markers including sCD14. Circulating levels of LBP and soluble CD14 are recognized as clinical biomarkers of endotoxemia, and thus it is expected that it will be informative for understanding the severity of sepsis and septic shock.
Comment 5:
Please incorporate the reference for " several studies have reported......and obesity (Line 347-348).
Response:
We appreciate your suggestion on relevant literatures. We have added the following references in addition to Cani’s paper (Diabetes 2007 56, 1761-72)
- Festi, D.; Schiumerini, R.; Eusebi, L.H.; Marasco, G.; Taddia , M.; Colecchia, A.; Gut microbiota and metabolic syndrome. World J Gastroenterol. 2014, 20, 16079-94.
- Boroni Moreira, AP.; de Cássia Gonçalves Alfenas, R. The influence of endotoxemia on the molecular mechanisms of insulin resistance. Nutr Hosp. 2012, 27, 382-90.
We hope that our responses we provide here satisfactorily address all the issues and concerns you have noted.
Thank you for your valuable time and consideration.
Best regards,
Hiroshi
Hiroshi Tamura, Ph.D.
Reviewer 2 Report
There is a need for improvement on some of the flow in this manuscript. Firstly, the reviewer find a lack of graphical illustration rather perplexing in a review that describes the LAL technology. Secondly, it may be necessary to increase on the figures as having only a single figure in the entire review does not appear aesthetically appealing to readers, especially for those wanting to get quick information from glimpsing. Finally, section 7A and 7B seems rather confusing and these should either be incorporated better into the review without specific number or expanded in context. It is also necessary to reduce wordings like "great breakthrough" or "immense attention" or "expectionally helpful" as these only reflect poorly on the language usage in an academic discourse. It would certainly be useful to engage a native speaker to work on the flow of the text.
Author Response
Dear reviewer #2:
We appreciate your valuable comments that helped improve the manuscript substantially.
Our point-by-point responses are as follows.
Comment 1:
Firstly, the reviewer find a lack of graphical illustration rather perplexing in a review that describes the LAL technology.
Response:
Thank you very much for your beneficial suggestion. According to your advice, we have supplemented the section 7 with an illustration (Figure 2, see attached) to briefly describe the characteristics of LAL and non-LAL technologies in an easy-to-understand manner.
Comment 2:
Secondly, it may be necessary to increase on the figures as having only a single figure in the entire review does not appear aesthetically appealing to readers, especially for those wanting to get quick information from glimpsing.
Response:
We really appreciate your perspective. It looks like that Table 1 is busy and somewhat confusing. Therefore, it may be a little hard to figure out how to differentiate the features of various methods and techniques. In addition to above Figure 2, we have considerably revised Table 1 in consideration of style and effective use of color to help clearly understand entire technologies highlighted in the review. Table 2 was also revised. Please see the attached file.
Comment 3:
Finally, section 7A and 7B seems rather confusing and these should either be incorporated better into the review without specific number or expanded in context. It is also necessary to reduce wordings like "great breakthrough" or "immense attention" or "expectionally helpful" as these only reflect poorly on the language usage in an academic discourse.
Response:
Thank you for your thoughtful comments and advice. According to your suggestion, we have significantly revised the section 7 while deleting specific number. Please see below.
1. Section 7: As shown in Figure 2, endotoxin assay can be divided into two categories: LAL and non-LAL assay. The LAL assay is officially used with a different type of formulation that comprises conventional or endotoxin-specific reagents for both end-point and kinetic assay formats [6, 39]. Other techniques (modified LAL) include ESP, the bioluminescence assay using mutant luciferase, electrochemical LAL assay and the ELISA-like assay, as described earlier [34-37]. As a different approach, a lab-on-a-chip device (Endosafe-PTS, Charles River) was first introduced as a modified technique to further improve the usability and simplicity of the LAL assay [40]. Furthermore, LAL alternatives based on recombinant technologies have recently been attracting a great deal of attention from the perspective of the global pharmacopoeia [41-44]. Recombinant alternatives are specific to endotoxin, and consist of two types of rea-gents: recombinant Factor C and cascade enzymes (see “next-generation LAL technology”). Recently, Bolden et al. reviewed currently availa-ble recombinant alternatives to horseshoe crab blood lysates and their comparability [45]. Among non-LAL based assays, GC/MS determination aims to target lipid A and its 3-hydroxy fatty acid molecules. Some studies suggest that GC/MS analysis could be clinically effective if its sensitivity and accuracy were markedly improved [9, 46]. Besides, a quantitative method was developed by fluorescence labeling of 3-hydroxy fatty acid, but complex process with preparative HPLC is required (47) ELISA is a simple and specific method and could be used as a diagnostic tool if its sensitivity were significantly enhanced and if the system were able to recognize various LPS molecules [48]. For GC/MS and ELISA, the technical dif-ficulties in making them suitable for practical use in the laboratory do not appear to be easy to overcome. LPS capture methods can offer a beneficial approach for horseshoe crab conservation, while these depend on the extent to which LPS binding molecules recognize diverse structures of LPS [49]. LPS O-antigen has been successfully targeted to detect LPS using polyclonal and monoclonal antibodies against the O-antigen. However, it has often yielded results that are inconsistent with those of the LAL assay [50].
2. The revisions in other sections (8, 9 ,10) were made in the same way.
3. In addition, "great breakthrough", "immense attention" and "expectionally helpful" were changed to " breakthrough" , "attention" and " helpful".
English language and style
(x) Extensive editing of English language and style required
Response:
We submitted for expert English editing provided by MDPI, and have received a revised manuscript.(English editing: english-29393) We are very satisfied with MDPI’s high-quality editing work.
Again, thank you very much for giving us the opportunity to strengthen our manuscript with your valuable comments and suggestions. We have worked hard to incorporate your feedback and really hope that the revision persuade you to accept our paper.
Thank you for your precious time and consideration.
Best regards,
Hiroshi
Hiroshi Tamura, Ph.D.
